# Determinant Factors of Achievement Motivation in School Physical Education

**DOI:** 10.3390/children9091366

**Published:** 2022-09-08

**Authors:** Juan M. García-Ceberino, Sebastián Feu, María G. Gamero, Sergio J. Ibáñez

**Affiliations:** 1Facultad de Educación y Psicología, Universidad de Extremadura, Avenida de Elvas s/n, 06006 Badajoz, Spain; 2Emotion Research Group, University of Huelva, 21071 Huelva, Spain; 3Optimization of Training and Sports Performance Research Group (GOERD), University of Extremadura, 10003 Cáceres, Spain; 4Faculty of Sports Science, University of Extremadura, 10003 Cáceres, Spain

**Keywords:** experience, perceived motor competence, primary education, sex, teaching methodology

## Abstract

Today, it is important for physical education teachers to know students’ motivation profiles for learning. Therefore, this study aimed to analyze achievement motivation according to four variables: students’ sex, the taught sport modality, students’ experience and teaching methodology. Likewise, the effects of students’ sex and experience on the methodologies applied were analyzed. A total of 108 primary education students (10.95 ± 0.48 years), 54 boys and 54 girls, from three state schools participated in the study. The students answered the Achievement Motivation in Physical Education test (Spanish version) after participating in soccer and basketball programs based on different methodologies. Each class-group received a different educational program (soccer or basketball). The differences between the categories of the variables analyzed were calculated for each dimension of the motivational test through the Mann–Whitney U and Kruskal–Wallis H tests. The effects of students’ sex and experience on the teaching methodologies applied were analyzed using the univariate General Linear Model test. In soccer and basketball, boys perceived being more motor competent (*U* = 732.00; *p* < 0.05; *r* = 0.43) than girls. In addition, experienced students in both sports perceived being more motor competent (*U* = 695.50; *p* < 0.05; *r* = 0.27) than inexperienced students. In turn, they indicated feeling less failure anxiety and stress (*U* = 780.00; *p* < 0.05; *r* = 0.22). All of the students who played soccer reported more commitment (learning dedication) (*U* = 1051.50; *p* < 0.05; *r* = 0.20) and perceived motor competence (*U* = 972.00; *p* < 0.05; *r* = 0.24) than students who played basketball. Considering the effects of students’ sex and experience on the methodologies (perceived motor competence dimension), there were significant differences (*F* = 7.68; *p* < 0.05; *ηp*^2^ = 0.07) in favor of experienced boys who played soccer and basketball using the Tactical Games Approach methodology. Soccer was practiced more in school and out of school. This made students feel greater commitment (learning dedication) and perceived motor competence towards this invasion sport in physical education. In addition, it was shown that teachers should take into account students’ sex and experience because they are two factors that influence the teaching of sports and achievement motivation.

## 1. Introduction

Physical activity offers benefits for: long-term memory and learning efficiency [1]; attention and motivation processes [2]; and mood, reducing stress that negatively affects the learning process [3]. For these reasons, one of the current purposes of physical education (PE) is to make students aware of the importance and value of regular physical activity and acquiring healthy lifestyle habits (adherence to physical activity) [4]. In this regard, in order to increase levels of physical activity in school and out-of-school contexts, students must be motivated [5,6] and feel competent [7].

Motivation is defined as the set of external (e.g., a prize) and internal (e.g., fun) factors that determine the behavior of individuals and that affect the choice, maintenance or abandonment of any activity [8]. Pintrich and Schunk [9] mentioned three constructs relevant to motivation in school contexts: academic self-efficacy, achievement motivation and causal attribution of achievement. The first construct refers to the students’ thinking about their ability to perform learning tasks proposed by the teacher. The second construct refers to students’ objectives and their thinking about the importance and interest in achieving them. Finally, the third construct refers to the consequences resulting from the performance of learning tasks, as well as the results of success or failure at the academic level. Research in recent years has focused on achievement motivation [10,11].

Achievement motivation in students learning PE is based on McClelland’s needs theory (the need for affiliation, the need for power and the need for achievement goals for conflict resolution) and, more precisely, on Atkinson’s theory of achievement goal motivation [12,13]. According to the latter theory, students base their motivation on the behavior of striving for success in an effort to achieve a better performance within an ideal standard of achievement and therefore reduce the tendency to fail in the learning task [12]. Students manifest a generic motivation towards the achievement of learning in PE, which encompasses their emotional disposition, the awareness of the value of what they are going to learn, the self-evaluation of their own competencies and the effect of their past experiences [14]. Therefore, students’ motivation state is influenced by the following factors: diligence in learning, valuation of learning, perceived motor competence and the fear of making mistakes [15]. Previous studies have analyzed achievement motivation in educational PE. Sánchez-Alcaraz et al. [10] analyzed achievement motivation as a function of students’ sex and educational grade (primary and secondary education). Martin-Moya et al. [11] identified motivational variations in high school after applying a gamification program on healthy habits and physical activity called DiverHealth. As in this study, Márquez-Barquero et al. [13] analyzed factors associated with achievement motivation, such as commitment to and involvement in learning, perceived motor competence and error anxiety during PE classes.

The teaching methodology used by teachers also determines the level of students’ motivation [16]. In general, teachers are used to implementing teaching units based on the traditional method of Direct Instruction (DI) [17,18]. In this methodology, the teacher is the main protagonist of the teaching–learning process and proposes learning tasks to develop movement patterns that the student must acquire through repetition, which causes little cognitive and motor involvement [18]. Teaching is based on the acquisition of individual technical skills through analytical situations, which are later incorporated into the real game when automation already exists [19]. The teacher uses prescriptive feedback in order to correct errors [18]. On the other hand, the major benefits of innovative and active methodologies, e.g., the Tactical Games Approach (TGA) method [20], have resulted in their increasing use in PE. TGA teaching is based on the game and promotes the cognitive involvement of the student, provoking, from the design of the learning tasks and the feedback used, the resolution of tactical problems [21]. The teacher proposes a tactical problem that is exaggerated by modifying the rules, contextualizing the task in the real game and thus promoting student learning [20]. Interrogative feedback is used, seeking student reflection to achieve meaningful learning [22]. The active role that the TGA method gives to the students facilitates an increase in students’ motivation in the construction of their own learning [16]. This change in methodological conception is slow, and there is still resistance to use this active and innovative methodology [23]. For this reason, both teaching methodologies have been selected, with the aim of finding the motivational differences provoked by their real practical application in the school context.

Some researchers have already applied the TGA methodology in the school context and have focused their interest on knowing some of the factors involved in the sports teaching process, such as: technical and tactical learning in situations played [24,25], the knowledge acquired [26,27] and the physical–physiological demands generated by this methodology [28,29,30]. Another study [31] analyzed the pedagogical variables and subjective external intensities of a basketball teaching unit designed and taught by an in-service PE teacher (Service Teacher’s Basketball unit, STBU) in order to determine which methodology, DI or TGA, it approximates. Furthermore, the scientific literature indicates that primary students’ sex and experience are factors that affect the study of methodologies and psychological parameters when they play soccer [7,25,26] and basketball [27,28].

In short, to achieve greater motivation for learning, the contents must have maximum meaning and significance for students and must be appropriate to their needs and motivational interests [32]. Teachers need to understand how motivation manifests itself in school PE. Therefore, this study aimed to analyze achievement motivation according to four factors: students’ sex, the taught sport modality, students’ experience and teaching methodology. The effects of the student’s sex and experience on the teaching methodologies applied were also analyzed. We hypothesized that: (1) Experienced boys who play soccer using the TGA methodology will report higher achievement motivation levels; and (2) Students’ sex and experience will influence the teaching methodologies applied.

## 2. Materials and Methods

### 2.1. Study Design

A quasi-experimental cross-sectional study was conducted [33] because the aim was to compare groups of students at a given time after receiving instruction in soccer [34] and basketball [35].

### 2.2. Participants and Procedure

A total of 108 students (54 boys and 54 girls) from primary education participated in the study. They ranged in age from 10 to 12 years old and belonged to three state schools in southwestern Spain. The selection of participants was intentional and based on criteria of proximity and accessibility. That is, several state schools were selected that were close to the researchers’ place of residence and whose school administration agreed to participate in the study with fifth- and sixth-grade students.

The characteristics of the students by school are shown in Table 1.

Despite the fact that the study did not require invasive measures to obtain the data, the University Bioethics Committee was asked for its approval [protocol code: 105/2022]. Authorization was also requested from the three schools and PE teachers. Once these authorizations were obtained, the parents or legal guardians were required to sign an informed consent form. Likewise, the study was approved by the school council within the school curriculum.

The intervention then began. Each class-group was randomly instructed on an educational program, soccer or basketball, according to different teaching methodologies (Figure 1). The class-groups were not modified to maintain the study’s ecological validity. The instruction of the educational programs lasted between two and three months in each school. Two PE teachers (and study researchers) taught the DI and TGA classes, and they had experience in planning and teaching both methodologies. Content validity and internal consistency were calculated using Aiken’s V and Cronbach’s Alpha coefficients, respectively, which indicated that DI and TGA programs are valid and reliable [36,37]. An in-service PE teacher, with a permanent position at the school, taught the STBU program.

After participating in the soccer [34] and basketball [35] educational programs based on different methodologies, the students completed the Achievement Motivation in Physical Education test based on Atkinson’s theory [14], adapted to Spanish (AMPET-e) [15], during regular school hours. The researchers were present in the classroom to address possible concerns and guarantee the anonymity of the answers. None of the students had problems answering the questions in the instrument. Inclusion criteria were: (1) participating in at least 80% of the soccer and basketball sessions; and (2) answering all of the AMPET-e instrument items.

The AMPET-e instrument [15] aims to ascertain how students think, feel and experience the situations presented to them in PE classes. It is composed of 37 items that measure three dimensions. Responses are given on a Likert-type scale from 1 to 5 points, where 1 means strongly disagree and 5 means strongly agree. The dimensions, considered the dependent variables of the study, were:(1).Commitment and dedication to learning (positive dimension): it consists of 15 items that measure the subjects’ emotional disposition and awareness of the value of what is to be learned.(2).Perceived motor competence (positive dimension): it is composed of 7 items that measure the subjects’ perceived ability and competence, as well as confidence in their physical fitness.(3).Failure anxiety and stress (negative dimension): it consists of 15 items that measure excessive tension that manifests itself in demanding and achievement situations, as well as lack of confidence.

The order of the scores in the failure anxiety and stress dimension is reversed; i.e., a higher score means less anxiety and stress. This procedure has been used in the literature [38]. Therefore, 5 means the most positive value, and 1 means the most negative value in the three dimensions. Then, the fit of the factorial structure of the original scale (of the AMPET-e) proposed by the authors [15] and the reliability of the instrument were assessed (Outcomes section).

Finally, the statistical analysis of the data was performed. The demographic and sports variables, established as independent variables to find differences between groups, were: the students’ sex (boys/girls), the taught sport modality (soccer/basketball) and the students’ experience in soccer or basketball (yes/no). Furthermore, the teaching method (TGA/DI/STBU) was used for group comparison. The purpose of using the STBU program [31] was to determine the achievement motivation generated by a teaching unit designed and taught by an in-service PE teacher. The rest of the educational programs have been intentionally designed under specific teaching methodologies.

Likewise, the study protocol respected the ethical guidelines of the Helsinki Declaration of 1975 (with modifications in subsequent years) and the Organic Law 3/2018 of December 5 on the protection of personal research data and the guarantee of digital rights (BOE, 294, 6 December 2018) to guarantee the ethical considerations of scientific research with human beings.

### 2.3. Outcomes

A Confirmatory Factor Analysis (CFA) of the AMPET-e instrument [15] was performed to assess the model’s goodness of fit. The following fit indices were used: (a) the Chi-Square test of model fit, where non-significant chi-square values (*p* > 0.05) are required to obtain a good model fit; (b) the Chi-Square ratio over Degrees of Freedom (CMIN/DF), with values less than 3 indicating a good fit; (c) the Root Mean Square Error of Approximation (RMSEA), where values below 0.05 are excellent, values between 0.05–0.08 are good and values above 0.08 are mediocre; (d) the Comparative Fit Index (CFI); (e) the Tucker Lewis Index (TLI) or Non-Normed Fit Index (NNFI); and (f) Incremental Fit Index (IFI), where values greater than 0.90 indicate an acceptable fit, and those greater than 0.95 indicate an excellent fit for the CFI, TLI/NNFI and IFI [39]. The AMOS plugin (for SPSS 25.0 statistical software) was used for the CFA [40].

The CFA indicated the removal of items from the motivational instrument to obtain an excellent model fit. By eliminating these items and with 4 items per dimension, a relation was established for 10 subjects/items (Table 2).

Likewise, reliability was calculated using the Average Variance Extracted (AVE) and Composite Reliability (CR) (Table 2). Hair et al. [41] stated an AVE ≥ 0.50 on all indicators and a CR ≥ 0.70 as requirements. Cronbach’s Alpha coefficient [42] showed the following reliability values: AMPET-e *α* = 0.72, moderate; commitment and dedication to learning *α* = 0.68, low; perceived motor competence *α* = 0.83, adequate; and failure anxiety and stress environment *α* = 0.75, moderate. In this regard, the literature suggests that 0.60 is the minimum acceptable value for Cronbach’s Alpha coefficient [41].

The instrument shows an excellent model fit and adequate reliability; therefore, the data from the study are relevant to the literature.

### 2.4. Statistical Analysis

Firstly, the criterion assumption tests (K-S, Rachas and Levene tests) were conducted to identify the hypothesis-testing model. Non-parametric mathematical tests were used for comparison [43].

Then, descriptive results were calculated as the mean and standard deviation. For each of the dimensions, the Mann–Whitney U (2 categories) and Kruskal–Wallis H (>2 categories) statistical tests were also calculated to analyze the differences (significance level, *p* < 0.05) between the categories of the independent variables studied [43].

Subsequently, these dimensions were transformed to use parametric mathematical tests [44] in order to analyze the effects of students’ sex and experience on the teaching methods applied. The univariate General Linear Model (univariate GLM) test was used [43].

Finally, the effect size was calculated using Rosenthal’s r formula (for the Mann–Whitney U test), the Epsilon-Squared (*E*^2^*_R_*) coefficient (for the Kruskal–Wallis H test) and the Partial Eta-Squared (*ηp*^2^) index (for the univariate GLM test) [45,46].

SPSS 25.0 statistical software (IBM Corp. Released 2017. IBM SPSS Statistics for Windows, Version 25, IBM Corp, Armonk, NY, USA) was used for the statistical analyses. Likewise, graphics were created with GraphPad Prism v8.0.1 (Graphpad Inc., La Jolla, CA, USA, EE. UU).

## 3. Results

The descriptive results of the three dimensions according to the comparison variables are shown in Figure 2. The value 5 means a positive value in all dimensions.

The differences between the different categories of each of the comparison variables by dimension are shown in Table 3. In terms of students’ sex, boys perceived being more motor competent than girls in soccer and basketball sports. In addition, experienced students in both sports perceived being more motor competent than inexperienced students. Likewise, they indicated feeling less failure anxiety and stress.

The students (boys and girls as a whole) who played soccer showed more commitment (dedication to learning) and perceived motor competence than students who played basketball.

There were no significant differences according to the teaching methodology applied (Table 3). Therefore, the possible effects of students’ sex and experience on these methodologies (TGA, DI and STBU) were studied (Table 4). In this regard, in the perceived motor competence dimension, there were no differences in the interaction teaching methodology*students’ experience (*F* = 0.99; *p* > 0.05; *ηp*^2^ = 0.02). When the variables teaching methodology*students’ sex were analyzed, there were significant differences showing that sex affected the study of the methodologies (*F* = 3.21; *p* < 0.05; *ηp*^2^ = 0.06). Likewise, as shown in the following table, the multiple interaction methodology*students’ sex*students’ experience only caused significant differences in the perceived motor competence dimension.

## 4. Discussion

PE teachers must be aware of the achievement motivation that their classes generate in order to improve the quality of student learning. Therefore, this study analyzed achievement motivation according to the following factors: students’ sex, the taught sport modality, students’ experience and teaching methodology. The effects of the student’s sex and experience on the methods used was also analyzed. The main findings show that boys perceived being more motor competent than girls in soccer and basketball. Furthermore, experienced students in both sports perceived being more motor competent than inexperienced students. In turn, they indicated feeling less failure anxiety and stress. The students who played soccer reported more commitment (learning dedication) and perceived motor competence than students who played basketball. Considering the effects of students’ sex and experience on the teaching methodologies, there were significant differences only in the dimension of perceived motor competence in favor of experienced boys who played soccer and basketball using the TGA methodology.

In soccer and basketball, boys showed higher values in the commitment and dedication to learning (non-significant) and perceived motor competence (significant) dimensions than girls. These results are in accordance with those reported in the study by Sánchez-Alcaraz et al. [10] when they analyzed achievement motivation and motivational orientation in primary and secondary school students. In contrast to the present study, these authors recorded higher failure anxiety and stress in girls. Furthermore, they stated that anxiety correlated positively with commitment and dedication to the sport and negatively with perceived competence. Márquez-Barquero et al. [13] also reported higher anxiety and stress in girls in PE classes, and they noted that positive reinforcement should be used to reduce these anxiety and stress levels by focusing attention on girls. In turn, García-Ceberino et al. [7] reported boys had a greater increase in perceived motor competence when playing soccer in primary PE, which was associated with a higher adherence degree in other sports contexts. Moreno-Murcia et al. [47] also obtained similar results in favor of the male sex in terms of perceived competence. There is a positive relationship between the intention to practice a sport and motivation [48]. Therefore, the level of physical activity is higher in boys than in girls [49].

It has been shown that differences according to students’ sex can be affected by other factors, such as: the sport level of the students, the task type, the presence of competition and the feedback type from the teacher [10]. Thus, when examining students’ experience in soccer and basketball, experienced students show higher values in the perceived motor competence dimension than inexperienced students. Likewise, greater perceived motor competence means less failure anxiety and stress. A previous study reported that greater competitive experience also implies lower anxiety [50]. The students tend to value their own abilities and their learning process more than those of their peers. In this regard, the sports experience provides students with the possibility of perceiving their ability as superior to that of others [7]. In contrast to this, Lamoneda and Huertas-Delgado [51] reported that the students analyzed in their study who did not practice a federated sport reported greater competence.

The students (boys and girls as a whole) who played soccer showed more commitment (dedication to learning) and perceived motor competence than students who played basketball. This could be due to the fact that soccer is the most practiced sport in Spain at school age [52]. In contrast to this study, a previous study on motivational variables in the out-of-school context reported that school-age basketball players had better motivational values, e.g., intrinsic motivation toward achievement when the training sessions are more task-oriented than ego-oriented, relative to soccer players [53]. In the same context, it has been shown that athletes who play team sports have higher motivation levels than those who play individual sports [54]. Therefore, teaching soccer and basketball in PE could improve students’ adherence to these sports in other contexts, e.g., clubs or public institutions.

There were no significant differences in any of the dimensions according to the teaching method. This could be due to the fact that PE already implies motivation in students. In this regard, González-Espinosa [16] also reported that there were no significant differences in the motivation of the primary students between DI and TGA methodologies after learning basketball. Although there were no differences between the two methodologies, this author recommends the employment of an active methodology because it improves intrinsic, identified and introjected regulation compared to the traditional methodology. The use of active methodologies for the game and the variability of new learning tasks are fundamental to achieving greater motivation in PE [11,55].

In contrast, there were significant differences in the perceived motor competence dimension when analyzing the multiple interaction methodology*students’ sex*students’ experience in favor of experienced boys who played soccer and basketball using the TGA method. In PE, Flores et al. [56] related gamified active methodologies to a reduction in anxiety and oppression in the face of failure in primary school girls. Students’ sex and experience in interacting with DI and TGA methods have also affected other factors in school soccer and basketball, such as: learning technical skills and tactical awareness [24,25]; declarative and procedural knowledge [26,27]; external and internal intensities [28,29]; and the degree of adherence to the sport [7].

## 5. Strengths and Limitations

Motivation for learning in PE should be an important task of school sport psychology. There is a need to improve motivation in school sports, especially intrinsic motivation [54], in order to increase physical activity levels. It is up to sports professionals to propose effective intervention strategies for sports practice [13]. Thus, the present study provides very useful information for PE teachers to promote their achievement motivation strategies more effectively towards their students’ sports practice.

Finally, it is necessary to state the limitations of the study. The sample size needs to be increased, and there is a need to study other factors that have been related to motivation, such as the perception of effort, enjoyment, sport adherence and/or learning. Future studies should also collect pre-test data to analyze intra-group differences and measure task-oriented and ego-oriented achievement motivation.

## 6. Conclusions

Students should be motivated in PE to increase levels of physical activity in school and out-of-school contexts. The study results show that boys perceived being more motor competent than girls in soccer and basketball. Furthermore, students with experience in both sports perceived being more motor competent than inexperienced students. Feeling more competent also made them feel less failure anxiety and stress. Considering the taught sport modality, students who played soccer, a more socially popular sport, showed more commitment (dedication to learning) and perceived motor competence than students who played basketball. In addition, PE classes are mixed and heterogeneous in the Spanish educational system. Therefore, based on our results, PE teachers should pay attention to students’ sex and experience when teaching invasion sports such as soccer and basketball, since both factors interact significantly with the teaching method. Knowing effective achievement motivation strategies will promote students’ sports adherence, both in and out of school, and therefore, health benefits.

## Figures and Tables

**Figure 1 children-09-01366-f001:**
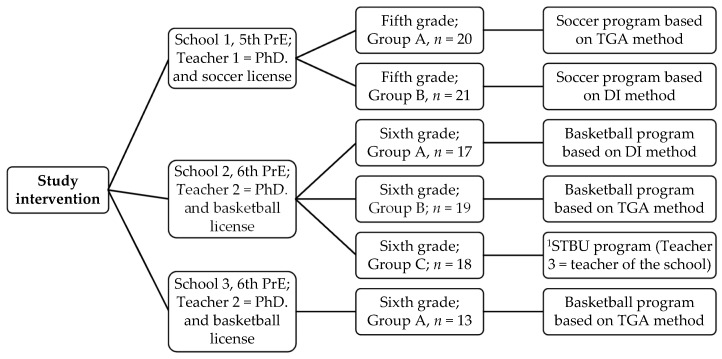
Distribution of student groups by school and educational program. Note: PrE = Primary Education; PhD = Doctorate in Physical Activity and Sport; TGA = Tactical Games Approach methodology; DI = Direct Instruction methodology; STBU = Service Teacher’s Basketball unit; ^1^ STBU is a teaching unit designed and taught by an in-service teacher.

**Figure 2 children-09-01366-f002:**
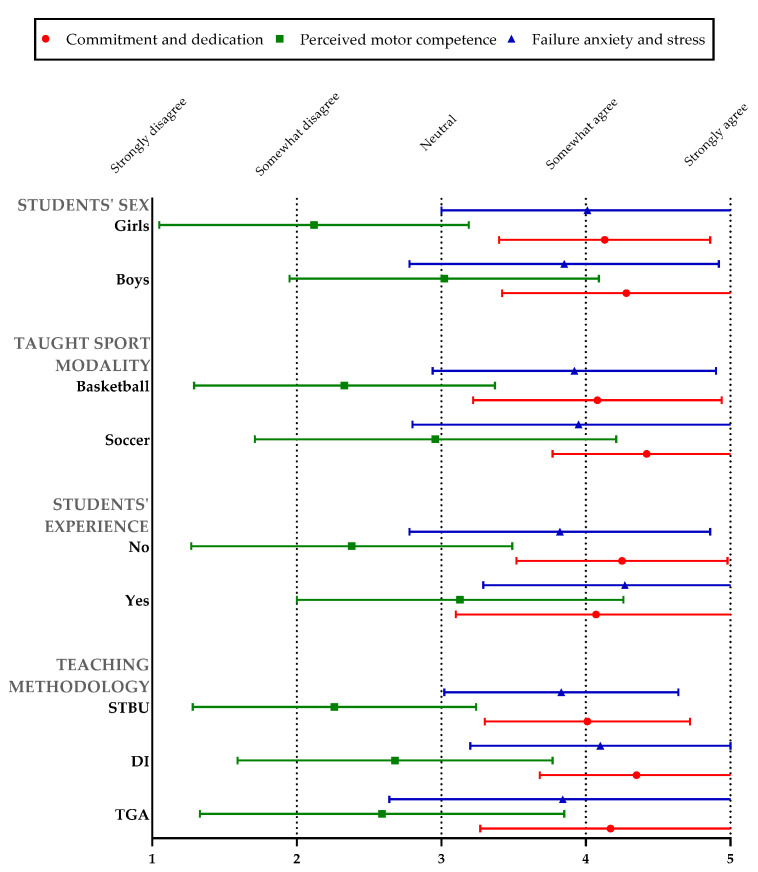
Descriptive results of the three dimensions according to the independent variables. Note: TGA = Tactical Games Approach methodology; DI = Direct Instruction methodology; STBU = Service Teacher´s Basketball unit.

**Table 1 children-09-01366-t001:** Characteristics of the primary education students participating.

Demographic and Sports Data	All	School 1, 5th PrE	School 2, 6th PrE	School 3, 6th PrE
Years (*M* ± *SD*)	10.95 ± 0.48	10.63 ± 0.49	11.09 ± 0.29	11.38 ± 0.51
Class-groups	6 class-groups	2 class-groups	3 class-groups	1 class-group
All students, girls	108, 54 girls	41, 18 girls	54, 31 girls	13, 5 girls
Taught sport modality	Both	Soccer	Basketball	Basketball
With experience	27 students	12 students	15 students	No students

Note: *M* = Mean; *SD* = Standard Deviation; PrE = Primary Education.

**Table 2 children-09-01366-t002:** CFA, AVE and CR of the AMPET-e instrument.

Model	*X* ^2^	CMIN/DF	RMSEA	CFI	TLI/NNFI	IFI	AVE	CR
Initial, 37 items	0.00	1.62	0.08	0.81	0.79	0.82	-	-
Modified, 12 items	0.07	1.31	0.05	0.96	0.94	0.96	0.65	0.83

Note: *X*^2^ = *p*-value of the Chi-Square test; CMIN/DF = Chi-Square ratio over Degrees of Freedom; RMSEA = Root Mean Square Error of Approximation; CFI = Comparative Fit Index; TLI = Tucker Lewis Index; NNFI = Non-Normed Fit Index; IFI = Incremental Fit Index; AVE = Average Variance Extracted; CR = Composite Reliability.

**Table 3 children-09-01366-t003:** Differences between the categories of each of the independent variables by dimension.

Variable	Dimension	*U*/*X*^2^	*p*	*r*/*E*^2^*_R_*	Comparison
Students’ sex	Commitment	1195.00	0.10	0.16	
	Perceived competence	732.00	0.00 *	0.43	Boys > Girls
	Anxiety and stress	1337.00	0.45	0.07	
Taught sport modality	Commitment	1051.50	0.04 *	0.20	Soccer > Basketball
	Perceived competence	972.00	0.01 *	0.24	Soccer > Basketball
	Anxiety and stress	1292.50	0.60	0.05	
Students’ experience	Commitment	1042.00	0.71	0.04	
	Perceived competence	695.50	0.01 *	0.27	Yes > No
	Anxiety and stress	780.00	0.03 *	0.22	Yes < No
^1^ Teaching methodology	Commitment	2.77	0.25	0.03	
	Perceived competence	1.66	0.44	0.02	
	Anxiety and stress	1.98	0.37	0.02	

Note: *U* = Mann–Whitney U test; *X*^2^ = Kruskal–Wallis H test; *r* = Rosenthal r formula; *E*^2^*_R_* = Epsilon-Squared coefficient; ^1^ Kruskal–Wallis U test was used (three categories); * *p* < 0.05.

**Table 4 children-09-01366-t004:** Effects of the students’ sex and experience on the methodologies applied (multiple interaction).

Dimension	Teaching Methodology	Students’ Sex	Students’ Experience	*M* ± *SD*	*F*	*p*	*ηp* ^2^
Commitment and dedication	TGA	Boys	Yes	1.92 ± 0.48	0.00	0.95	0.00
			No	2.08 ± 0.21			
		Girls	Yes	1.88 ± 0.17			
			No	2.00 ± 0.22			
	DI	Boys	Yes	2.06 ± 0.27			
			No	2.09 ± 0.15			
		Girls	Yes	2.08 ± 0.10			
			No	2.08 ± 0.13			
	STBU	Boys	Yes	-			
			No	2.00 ± 0.19			
		Girls	Yes	1.94 ± 0.24			
			No	2.03 ± 0.16			
Perceived motor competence	TGA	Boys	Yes	2.12 ± 0.17	7.68	0.01 *	0.07
			No	1.68 ± 0.34			
		Girls	Yes	1.37 ± 0.30			
			No	1.40 ± 0.34			
	DI	Boys	Yes	1.79 ± 0.25			
			No	1.63 ± 0.34			
		Girls	Yes	1.88 ± 0.25			
			No	1.30 ± 0.23			
	STBU	Boys	Yes	-			
			No	1.60 ± 0.19			
		Girls	Yes	1.41 ± 0.12			
			No	1.35 ± 0.47			
Failure anxiety and stress	TGA	Boys	Yes	1.85 ± 0.58	0.83	0.37	0.01
			No	1.92 ± 0.34			
		Girls	Yes	2.16 ± 0.11			
			No	1.91 ± 0.34			
	DI	Boys	Yes	2.06 ± 0.26			
			No	1.96 ± 0.20			
		Girls	Yes	2.12 ± 0.11			
			No	1.98 ± 0.28			
	STBU	Boys	Yes	-			
			No	1.86 ± 0.24			
		Girls	Yes	2.01 ± 0.17			
			No	2.01 ± 0.20			

Note: *M* = Mean; *SD* = Standard Deviation; *F* = Univariate General Linear Model test; *ηp*^2^ = Partial Eta-Squared index; TGA = Tactical Games Approach methodology; DI = Direct Instruction methodology; STBU = Service Teacher´s Basketball unit; * *p* < 0.05.

## Data Availability

Data will be available upon reasonable request to the corresponding author.

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
