# Peer review of "Determinant Factors of Achievement Motivation in School Physical Education"

_children, 2022, doi:10.3390/children9091366_

Round 1
Reviewer 1 Report
The authors describe their attempt to examine determinant factors of achievement-motivation in primary school physical education. In this claimed "quasi-experimental cross-sectional" study, 108 participants were assigned to a Directed Instruction (DI) or a Tactical Game Approach (TGA) group. The self-reported Achievement Motivation in Physical Education test was collected after the intervention. It was claimed that the achievement motivation was not different between the DI and the TGA in the data analysis.
The paper appears to be an interesting attempt to explore students' motivation in two different approaches, DI and TGA. However, many issues with the introduction, methods, and analysis need to be clarified/addressed. Furthermore, some conclusions overreach the data collected, while other significant results are given less emphasis than they warrant. Below are more specific comments by section:
1 Introduction
Motivation theories are nebulas in the psychology discipline. In lines 47 to 57, the authors provided some basic explanations about motivation and three constructs, self-efficacy, achievement motivation, and causal attribution of achievement. Then, this manuscript was narrowed down to the Achievement Motivation Theory (lines 58 to 65).
According to Lussier and Achua (2007, p. 42), “Achievement Motivation Theory attempts to explain and predict behavior and performance based on a person's need for achievement, power, and affiliation” (Lussier & Achua, 2007, p. 42). The Achievement Motivation Theory is also referred to as the Acquired Needs Theory or the Learned Needs Theory. There are four major theories in the need-based category: Maslow's hierarchy of needs, ERG theory, Herzberg's dual-factor theory, and McClelland's acquired needs theory. Which theories are you adopting in your definition? Please explain it in your manuscript.
I cannot match this definition with the authors' description in lines 58 to 65. In addition, the introduction did not provide enough previous research about the Achievement Motivation Theory (in physical education). The information provided by the authors was deficient. This caused the reviewer hard to understand the achievement motivation research in physical education from lines 75 to 79.
Overall, I understand that the authors try to compare two different teaching methods, Direct Instruction vs. Tactical Games Approach, to examine students' achievement motivation. However, I believe the author needs put extra effort and time into improving their introduction to make it clear and solid. Otherwise, the research question and the hypothesis are not substantially supported.
Additional questions in the introduction:
What is the definition of TGA?
Why are you using TGA in this study?
Any previous TGA research in PE?
2 Materials and Methods
How long of the intervention lasted in this study, weeks, months, or a semester?
How many teachers implemented the DI and TGA classes?
Does the teacher (s) have any experience in teaching TGA-based PE?
How do you assess the validity of both DI and TGA implementation?
Lines 92 to 95, are both groups received both soccer and basketball? Explain why they were treated in different sports.
Line 112 to 116 (figure 1), why does STBU appears here? Why is the STBU not mentioned in the introduction? What is the purpose of having this STBU group in the intervention?
Lines 117 to 119, do you have a pre-test with the Achievement Motivation in PE? Explain why you are not collect the pre-test (Achievement Motivation in PE) before the intervention?
Lines 125 to 138, what are the sample questions in each latent variable?
Lines 125, does the AMPET-e validated? Where is the original instrument comes from (reference)?
Line 155, where is the correlation matrix table? I am highly concerned with the correlation table (raw data) - please submit your correlation matrix table so the reviewer can confirm your results and analysis (lines 155 to 206).
3 Results
First, I have questions about combined perceived ability, perceived competence, and physical fitness confidence to one dimension - perceived motor competence. There are other validated instruments to measure perceived motor competency. The authors can use the motor competency instrument to measure their participants rather than shift one of the latent variables in achievement motivation to represent perceived motor competency. This is not appropriate.
In addition, the introduction did not elaborate enough on achievement motivation literature in physical education. The detachment between the introduction and the outcome (discussion) will cause this manuscript to lose its focal point.
4 Discussion
Numerous research in PE mentioned that boys are more physically competent than girls. And David Stodden's model (Quest, 2008) already substantiated that "children’s motor skill competence
plays a highly significant yet varying role in supporting physical activity behaviors" - which made this study lack new or originality IF motor competency is the focus of the discussion.
Repeatedly, the primary concern in this manuscript is that the author did poorly in their introduction, so this manuscript
- lacks new or original ideas about the achievement motivation,
- absence of an acceptable scientific rationale in their research question,
- lack of experience in the fundamental methodology,
- lack of sufficient experimental details,
- uncertainty concerning future directions.
Additional concerns about the discussion, (1) this study did not measure task, ego-oriented achievement motivation (and Intrinsic Motivation Instrument). It is not appropriate to compare different motivation instruments (variables) in the discussion section.
At the same time, this study failed to compare its results with previous AMPET studies. With this in mind, the discussion generates limited information about the current study and future directions in the achievement motivation in physical education.
Author Response
First of all, we would like to express our gratitude to reviewer 1 for the time in reviewing our manuscript and for providing us comments helpful to improve this manuscript quality. We have found suggestions very constructive and have answered their concerns.
--------------------
All manuscript
- All corrections were marked in red and/or change control.
- An experienced native translator reviewed the translation of the manuscript (translation certificate attached).
--------------------
Reviewer’ note: Motivation theories are nebulas in the psychology discipline. In lines 47 to 57, the authors provided some basic explanations about motivation and three constructs, self-efficacy, achievement motivation, and causal attribution of achievement. Then, this manuscript was narrowed down to the Achievement Motivation Theory (lines 58 to 65). According to Lussier and Achua (2007, p. 42), “Achievement Motivation Theory attempts to explain and predict behavior and performance based on a person's need for achievement, power, and affiliation” (Lussier & Achua, 2007, p. 42). The Achievement Motivation Theory is also referred to as the Acquired Needs Theory or the Learned Needs Theory. There are four major theories in the need-based category: Maslow's hierarchy of needs, ERG theory, Herzberg's dual-factor theory, and McClelland's acquired needs theory. Which theories are you adopting in your definition? Please explain it in your manuscript.
Authors’ response: Based on your question, the theories adopted in this study have been defined in relation to achievement motivation (line 57 to 62).
--------------------
Reviewer’ note: I cannot match this definition with the authors' description in lines 58 to 65. In addition, the introduction did not provide enough previous research about the Achievement Motivation Theory (in physical education). The information provided by the authors was deficient. This caused the reviewer hard to understand the achievement motivation research in physical education from lines 75 to 79.
Authors’ response: Thank you for your suggestion. Lines 75 to 79 have been changed from paragraph to paragraph to help the reader understand the manuscript. In addition, a similar study on achievement motivation has been cited (line 72 to 75).
--------------------
Reviewer’ note: Overall, I understand that the authors try to compare two different teaching methods, Direct Instruction vs. Tactical Games Approach, to examine students' achievement motivation. However, I believe the author needs put extra effort and time into improving their introduction to make it clear and solid. Otherwise, the research question and the hypothesis are not substantially supported.
Authors’ response: I agree with your comment. In general, the introduction has been expanded in order to be clearer and more robust.
--------------------
Reviewer’ note: Additional questions in the introduction: What is the definition of TGA? Why are you using TGA in this study? Any previous TGA research in PE?
Authors’ response: The introduction has been expanded, the definition of TGA has been answered and the definition of DI has also been made to clarify the differences between the two (line 76 to 94).
In addition, the reason why we have used the TGA in the study has been explained. Finally, some of the studies that have used this methodology in the school context have been cited (line 94 to 104).
--------------------
Reviewer’ note: How long of the intervention lasted in this study, weeks, months, or a semester?
Authors’ response: Thank you for your suggestion. I have added the duration of the intervention (line 139 to 140).
--------------------
Reviewer’ note: How many teachers implemented the DI and TGA classes? Does the teacher (s) have any experience in teaching TGA-based PE?
Authors’ response: Based in your questions, the number of physical education teachers teaching DI and TGA classes has been indicated, as well as their experience in teaching both methodologies (line 140 to 141).
--------------------
Reviewer’ note: How do you assess the validity of both DI and TGA implementation?
Authors’ response: Thank you for your question. The coefficients used to calculate the validity and reliability of educational programs have been indicated (line 142 to 143).
--------------------
Reviewer’ note: Lines 92 to 95, are both groups received both soccer and basketball? Explain why they were treated in different sports
Authors’ response: The paragraph has been reworded (line 119 to 120). Likewise, Figure 1 shows the educational program that each class-group received.
--------------------
Reviewer’ note: Line 112 to 116 (figure 1), why does STBU appears here? Why is the STBU not mentioned in the introduction? What is the purpose of having this STBU group in the intervention?
Authors’ response: We agree with your comment. Information about the STBU program has been added to the introduction (line 101 to 104). Likewise, in Figure 1 it has been indicated which educational program is taught by each of the three physical education teachers. The purpose of using this program is to know the motivation generated by a teaching unit (without a specific methodology) designed and taught by an in-service teacher
--------------------
Reviewer’ note: Lines 117 to 119, do you have a pre-test with the Achievement Motivation in PE? Explain why you are not collect the pre-test (Achievement Motivation in PE) before the intervention?
Authors’ response: The main purpose of the study was to analyze and compare achievement motivation in primary school students after receiving different educational programs (inter-group differences). However, this suggestion has been added as a proposal for improvement to analyze intra-group differences (line 352).
--------------------
Reviewer’ note: Line 125, does the AMPET-e validated? Where is the original instrument comes from (reference)?
Authors’ response: Thank you for your suggestion. The reference of the original instrument has been added (line 152).
Nishida, T. Reliability and factor structure of the achievement motivation in physical education test. Journal of Sport and exercise Psychology 1988, 10, 418-430, doi:https://doi.org/10.1123/jsep.10.4.418.
--------------------
Reviewer’ note: Line 155, where is the correlation matrix table? I am highly concerned with the correlation table (raw data) - please submit your correlation matrix table so the reviewer can confirm your results and analysis (lines 155 to 206).
Authors’ response: Based in your comment, in Table 2, the goodness of fit of the initial model (37 items) has been added.
--------------------
Reviewer’ note: First, I have questions about combined perceived ability, perceived competence, and physical fitness confidence to one dimension - perceived motor competence. There are other validated instruments to measure perceived motor competency. The authors can use the motor competency instrument to measure their participants rather than shift one of the latent variables in achievement motivation to represent perceived motor competency. This is not appropriate.
Authors’ response: We disagree with your comment. We have used only the three dimensions recorded in the AMPET-e instrument (Spanish version). At no time have these dimensions been combined. Thus, the wording of the results section has been modified to avoid confusing the reader.
Ruiz, L.M.; Graupera, J.L.; Gutiérrez, M.; Nishida, T. El Test Ampet de motivación de logro para el aprendizaje en educación física: desarrollo y análisis factorial de la versión española. Revista de Educación 2004, 195-211.
--------------------
Reviewer’ note: In addition, the introduction did not elaborate enough on achievement motivation literature in physical education. The detachment between the introduction and the outcome (discussion) will cause this manuscript to lose its focal point.
Authors’ response: We agree with your comment. In the introduction and discussion sections, references on achievement motivation in physical education has been expanded.
Sánchez-Alcaraz, B.J.; Gómez-Mármol, A.; Más, M. Estudio de la motivación de logro y orientación motivacional en estudiantes de educación física. Apunts. Educación Física y Deportes 2016, 124, 35-40, doi:https://doi.org/10.5672/apunts.2014-0983.es.(2016/2).124.03.
Martín-Moya, R.; Ruiz-Montero, P.J.; Chiva-Bartoll, Ó.; Capella-Peris, C. Achievement motivation for learning in physical education students: Diverhealth. Revista Interamericana de Psicologia/Interamerican Journal of Psychology 2018, 52, 270-280.
Márquez-Barquero, M.; Azofeifa-Mora, C.; Rodríguez-Méndez, D. Factores de motivación de logro: el compromiso y entrega en el aprendizaje, la competencia motriz percibida, laansiedad ante el error y situaciones de estrés en estudiantes de cuarto, quinto y sexto nivel escolar durante laclase de educación física. Educación: Revista de la Universidad de Costa Rica 2019, 43, 61-72, doi:https://doi.org/10.15517/revedu.v43i1.33109.
Rodríguez, B.; Flores, G.; Fernández, J. Ansiedad ante el fracaso en educación física ¿puede la gamificación promover cambios en las alumnas de primaria? Retos. Nuevas Tendencias en Educación Física, Deporte y Recreación 2022, 44, 739-748.
--------------------
Reviewer’ note: Numerous research in PE mentioned that boys are more physically competent than girls. And David Stodden's model (Quest, 2008) already substantiated that "children’s motor skill competence plays a highly significant yet varying role in supporting physical activity behaviors" - which made this study lack new or originality IF motor competency is the focus of the discussion.
Authors’ response: We disagree with your comment. As mentioned above, this study focuses on three dimensions (of AMPET-e instrument): (1) commitment and dedication to learning; (2) perceived motor competence; and (3) failure anxiety and stress.
It reported interesting findings, such as that: (1) boys perceived being more motor competence than girls in soccer and basketball; (2) the experienced students in both sports perceived being more motor competence than inexperienced students. In turn, they indicated feeling less failure anxiety and stress; (3) the students who played soccer reported more commitment (learning dedication) and perceived motor competence than students who played basketball; and (4) considering the effect of students’ sex and experience on the teaching methodologies (perceived motor competence dimension), there were significant differences in favor of experienced boys who played soccer and basketball using the Tactical Games Approach methodology.
Therefore, it provides very useful information for PE teachers to promote their achievement motivation strategies more effectively towards their students' sport practice. For example, PE classes are mixed and heterogeneous in the Spanish educational system. Thus, based on our results, physical education teachers should pay attention to the students’ sex and experience when teaching invasion sports such as soccer and basketball, since both factors interact significantly with the teaching methods.
--------------------
Reviewer’ note: Repeatedly, the primary concern in this manuscript is that the author did poorly in their introduction.
Authors’ response: Thank you for your comment. The introduction section has been modified based on the reviewer's suggestions and now fits into the discussion.
--------------------
Reviewer’ note: Additional concerns about the discussion, (1) this study did not measure task, ego-oriented achievement motivation (and Intrinsic Motivation Instrument). It is not appropriate to compare different motivation instruments (variables) in the discussion section.
Authors’ response: Thank you for your suggestion. The dimensions recorded in the AMPET-e have only been studied in terms of several factors involved in the teaching-learning process in physical education classes. These results will help the teacher in his or her teaching work.
In addition, your suggestion has been added as a research perspective.
--------------------
Reviewer’ note: At the same time, this study failed to compare its results with previous AMPET studies. With this in mind, the discussion generates limited information about the current study and future directions in the achievement motivation in physical education.
Authors’ response: This study has discussed previous AMPET studies. Some studies were:
Sánchez-Alcaraz, B.J.; Gómez-Mármol, A.; Más, M. Estudio de la motivación de logro y orientación motivacional en estudiantes de educación física. Apunts. Educación Física y Deportes 2016, 124, 35-40, doi:https://doi.org/10.5672/apunts.2014-0983.es.(2016/2).124.03.
Martín-Moya, R.; Ruiz-Montero, P.J.; Chiva-Bartoll, Ó.; Capella-Peris, C. Achievement motivation for learning in physical education students: Diverhealth. Revista Interamericana de Psicologia/Interamerican Journal of Psychology 2018, 52, 270-280.
Márquez-Barquero, M.; Azofeifa-Mora, C.; Rodríguez-Méndez, D. Factores de motivación de logro: el compromiso y entrega en el aprendizaje, la competencia motriz percibida, laansiedad ante el error y situaciones de estrés en estudiantes de cuarto, quinto y sexto nivel escolar durante laclase de educación física. Educación: Revista de la Universidad de Costa Rica 2019, 43, 61-72, doi:https://doi.org/10.15517/revedu.v43i1.33109.
Rodríguez, B.; Flores, G.; Fernández, J. Ansiedad ante el fracaso en educación física ¿puede la gamificación promover cambios en las alumnas de primaria? Retos. Nuevas Tendencias en Educación Física, Deporte y Recreación 2022, 44, 739-748.

Reviewer 2 Report
Comment 1: Please state the main hypothesis of the research.
Comment 2: What is the sampling method? random sampling? Please describe more on how to select 108 students.
Comment 3: Please demonstrate the criteria of proximity and accessibility in detail.
Comment 4: The result section could be more concise. Some numbers can be found in the tables. Please try not to repeat them in the main text again unless they are especially important. The comment also applies to the rest of the result section.
Comment 5: For the discussion section, I suggest integrating the last two paragraphs (strengths and limitations)

Author Response
First of all, we would like to express our gratitude to reviewer 2 for the time in reviewing our manuscript and for providing us comments helpful to improve this manuscript quality. We have found suggestions very constructive and have answered their concerns.
--------------------
All manuscript
- All corrections were marked in red and/or change control.
- An experienced native translator reviewed the translation of the manuscript (translation certificate attached).
--------------------
Reviewer’ note: Please state the main hypothesis of the research.
Authors’ response: Thank you for your suggestion. The study hypotheses have been added (line 113 to 115).
--------------------
Reviewer’ note: What is the sampling method? Random sampling? Please describe more on how to select 108 students.
Authors’ response: Based on your questions, lines 124-125 indicate that participants were purposively selected based on proximity and accessibility criteria.
--------------------
Reviewer’ note: Please demonstrate the criteria of proximity and accessibility in detail.
Authors’ response: Based on your sugestion, the criteria of proximity and accessibility have been detailed (line 125 to 127).
--------------------
Reviewer’ note: The result section could be more concise. Some numbers can be found in the tables. Please try not to repeat them in the main text again unless they are especially important. The comment also applies to the rest of the result section.
Authors’ response: We consider that the results presented in the text are important and not repetitive. Table 4 shows the multiple interaction methodology*students’ sex*students’ experience; while in the text the interactions methodology*students’sex, and methodology*students’ experience are shown independently.
--------------------
Reviewer’ note: For the discussion section, I suggest integrating the last two paragraphs (strengths and limitations).
Authors’ response: Thank you for your comment. The last two paragraphs refer to the strengths and limitations of the study. In addition, research perspectives have been added (line 351 to 353).

Reviewer 3 Report
Congratulations to the authors for their work. It is an interesting study, but it is a study without a control group.
In the introduction, there is no argument to support the need to study the differences between two groups that undergo a basketball or football programme. The differences between the two sports are not discussed.
The sample is not well distributed and football is carried out by smaller pupils and in different schools.
The sample is not too large to make so many subgroups, which are less than 30.
The basketball programme uses a methodology that is not used in football, which may influence the comparisons depending on the educational methodology used.
The results obtained in the study do not offer a great contribution to knowledge. Mainly because the absence of significant differences in many of the study variables may be due to the small sample size of the subgroups.
Therefore, the conclusions are not very novel and are not strongly supported due to the limitations of the study.
Author Response
First of all, we would like to express our gratitude to reviewer 3 for the time in reviewing our manuscript and for providing us comments helpful to improve this manuscript quality. We have found suggestions very constructive and have answered their concerns.
--------------------
All manuscript
- All corrections were marked in red and/or change control.
- An experienced native translator reviewed the translation of the manuscript (translation certificate attached).
--------------------
Reviewer’ note: In the introduction, there is no argument to support the need to study the differences between two groups that undergo a basketball or football programme. The differences between the two sports are not discussed.
Authors’ response: I agree with your comments. In general, the introduction has been expanded in order to be clearer and more robust, and now fits into the discussion.
The differences between the two sports have been discussed (line 315 to 319).
--------------------
Reviewer’ note: The sample is not well distributed and football is carried out by smaller pupils and in different schools. The sample is not too large to make so many subgroups, which are less than 30.
Authors’ response: Thank you for your comments. The class-groups of both schools, mixed and heterogeneous, were already established by the Spanish educational system and were not modified to maintain the ecological validity of the study.
In addition, 5th and 6th grade students were selected because they are ready to address the technical skills and tactical awareness of soccer and basketball invasion sports (DOE, nº. 114, June 16, 2014).
DECRETO 103/2014, de 10 de junio, por el que se establece el currículo de Educación Primaria para la Comunidad Autónoma de Extremadura (DOE, nº. 114, 16 de junio de 2014).
--------------------
Reviewer’ note: The basketball programme uses a methodology that is not used in football, which may influence the comparisons depending on the educational methodology used.
Authors’ response: Thank you for your suggestion. Information about the STBU program has been added to the introduction (line 101 to 104).
The purpose of using this program was to know the motivation generated by a teaching unit (without a specific methodology, DI or TGA) designed and taught by an in-service teacher. A previous study has reported that the STBU program is approximate to the specific methodology of DI.
Gamero-Portillo, M.G.; García-Ceberino, J.M.; Rodriguez-Rocha, J.; Ibáñez, S.J.; Feu, S. Estudio de tres programas de intervención para la enseñanza del baloncesto en edad escolar. Un estudio de casos. E-balonamo Com 2022, 18, 127-148.
--------------------
Reviewer’ note: The results obtained in the study do not offer a great contribution to knowledge. Mainly because the absence of significant differences in many of the study variables may be due to the small sample size of the subgroups. Therefore, the conclusions are not very novel and are not strongly supported due to the limitations of the study.
Authors’ response: We disagree with your comments. This study reported interesting findings, such as that: (1) boys perceived being more motor competence than girls in soccer and basketball; (2) the experienced students in both sports perceived being more motor competence than inexperienced students. In turn, they indicated feeling less failure anxiety and stress; (3) the students who played soccer reported more commitment (learning dedication) and perceived motor competence than students who played basketball; and (4) considering the effect of students’ sex and experience on the teaching methodologies (perceived motor competence dimension), there were significant differences in favor of experienced boys who played soccer and basketball using the Tactical Games Approach methodology.
Therefore, it provides very useful information for PE teachers to promote their achievement motivation strategies more effectively towards their students' sport practice. For example, PE classes are mixed and heterogeneous in the Spanish educational system. Thus, based on our results, physical education teachers should pay attention to the students’ sex and experience when teaching invasion sports such as soccer and basketball, since both factors interact significantly with the teaching methods.
Furthermore, sample size was noted as a limitation of the study (line 349 to line 350). The conclusions are supported by the results of the study, which will help the teaching work.

Round 2
Reviewer 1 Report
N/A
Reviewer 2 Report
addressed my questions fully
Reviewer 3 Report
Congratulations to the authors for the improvements to the article.